# I-PHYRE: INTERACTIVE PHYSICAL REASONING

**Shiqian Li**[1,2]**, Kewen Wu**[2,3]**, Chi Zhang**[2✉]**, Yixin Zhu**[1✉]

[1] Institute for Artificial Intelligence, Peking University
[2] National Key Laboratory of General Artificial Intelligence, BIGAI
[3] Department of Automation, Tsinghua University

https://lishiqianhugh.github.io/IPHYRE/

## ABSTRACT

Current evaluation protocols predominantly assess physical reasoning in *stationary* scenes, creating a gap in evaluating agents' abilities to *interact* with dynamic events. While contemporary methods allow agents to modify initial scene configurations and observe consequences, they lack the capability to interact with events in real time. To address this, we introduce Interactive PHysical Reasoning (I-PHYRE), a framework that challenges agents to simultaneously exhibit intuitive physical reasoning, multi-step planning, and in-situ intervention. Here, *intuitive physical reasoning* refers to a quick, approximate understanding of physics to address complex problems; *multi-step* denotes the need for extensive planning sequences in I-PHYRE, considering each intervention can significantly alter subsequent choices; and *in-situ* implies the necessity for timely object manipulation within a scene, where minor timing deviations can result in task failure. We formulate four game splits to scrutinize agents' learning and generalization of essential principles of interactive physical reasoning, fostering learning through interaction with representative scenarios. Our exploration involves three planning strategies and examines several supervised and reinforcement agents' zero-shot generalization proficiency on I-PHYRE. The outcomes highlight a notable gap between existing learning algorithms and human performance, emphasizing the imperative for more research in equipping agents with interactive physical reasoning capabilities.

## 1 INTRODUCTION

Consider the dynamics of playing a game of 3D pinball, where achieving a high score is contingent upon making swift and accurate physical judgments. Players must meticulously control the launcher, anticipate the ball's trajectory, understand the paths it will take upon interacting with targets and bumpers, and determine the optimal moments to activate the flippers. Given the intricate nature of the ball's trajectories, players are compelled to adapt in real time to the game's sudden shifts; professional players demonstrate mastery in this game, with records of continuous play near three hours.[1] Indeed, such proficient and interactive physical reasoning is not exclusive to humans but a common trait observed across various species. Even toddlers exhibit a remarkable ability to interact seamlessly with the physical world, demonstrating capabilities in seeing, interacting, and in-situ planning (Spelke & Kinzler, 2007; Fazeli et al., 2019). Studies on animals such as crows (Gruber et al., 2019), pigeons (Epstein et al., 1984), and apes (Tecwyn et al., 2013) also reveal analogous multi-step planning abilities employed for solving complex problems.

The profound physical reasoning aptitude observed in humans and animals has spurred our interest in equipping intelligent agents with analogous capabilities. Contemporary benchmarks have emerged to assess the prowess of physical reasoning agents, ranging from determining the stability of a block stack (Lerer et al., 2016) to navigating intricate physical games (Bakhtin et al., 2019; Allen et al., 2020). However, these prior works often overlook a pivotal facet of physical reasoning—**interactivity**. This dimension demands that an agent concurrently: (i) understand the physical events **intuitively**, enabling swift yet approximate predictions of future outcomes, (ii) strategize **multi-step** actions, considering the anticipated environmental shifts resulting from interventions, and (iii) execute **in-situ** interventions, ensuring precise timing with the environment.

---

[1] https://www.youtube.com/watch?v=O4_848IPFVw

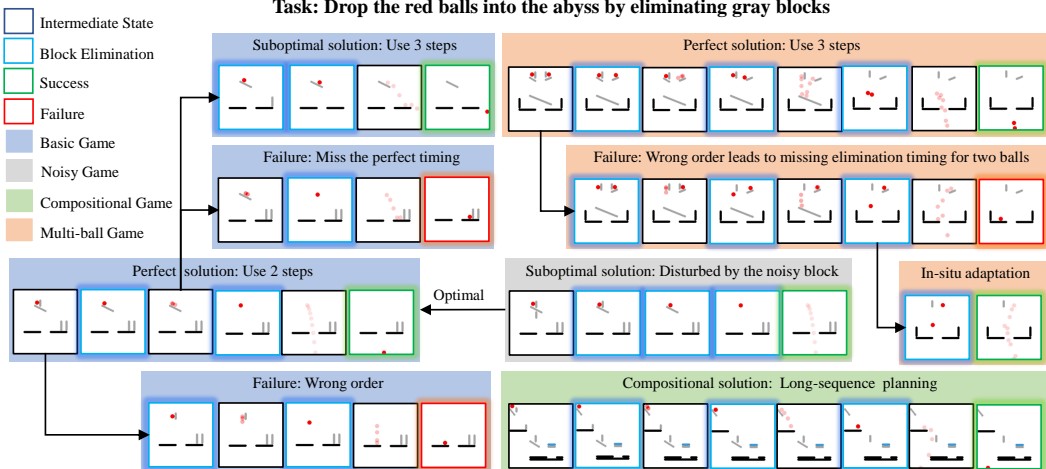

Figure 1: **Illustrations of the four splits in `I-PHYRE`.** The objective is to guide all red balls into the hole by strategically eliminating gray blocks in a stepwise manner. We showcase one game from each split (basic, noisy, compositional, and multi-ball) and explore the potential dynamics that different interventions could unfold. Images encased in a black border, cyan border, green border, and red border represent intermediate states, time steps at which an elimination action is undertaken, frames depicting success, and frames indicating failure, respectively. The gray and black blocks remain stationary, while the light blue blocks are mobile, influenced by gravity and collisions; only the gray blocks are subject to elimination. Successfully solving the games necessitates reasoning about the sequence of interventions and meticulously managing the exact timing of interventions.

Indeed, prevailing studies exhibit notable limitations in exploring physical reasoning due to the following constraints:

- Existing works predominantly permit either passive observation or a single-round intervention. These one-round settings encompass (i) rendering responses post-observation of dynamic scenarios (ee Baillargeon, 1987; Hespos & Baillargeon, 2001; Spelke et al., 1992), and (ii) modifying the initial setup through a singular intervention in a given stationary scene (Bakhtin et al., 2019; Allen et al., 2020). The restriction to a single action precludes the exploration of action compositionality in extended sequences and the sequential impact of physical interventions.
- Existing frameworks are inadequate for assessing the repercussions of action timing. Given that all interventions are executed while the scenes are static—albeit subsequent unfolding of dynamic events is possible (Bakhtin et al., 2019; Allen et al., 2020)—the evaluation of the temporal impact of actions remains unattainable.

To rectify the aforementioned shortcomings, we present Interactive PHysical Reasoning (`I-PHYRE`), the first benchmark explicitly crafted to bridge the existing gaps in physical reasoning. `I-PHYRE` necessitates an agent to contemplate the physical repercussions of its interventions and to interact sequentially with the environment with impeccable timing to attain the objective. `I-PHYRE` is conceptualized based on the following principles:

- **Intuitive physical reasoning**: Our focus is on fostering a rapid yet approximate capability for predicting physical outcomes, rather than an exact simulation or modeling of the physical events.
- **Multi-step interventions**: The resolution of a problem demands the execution of actions in multiple steps. In contrast to single-action setups, multi-step actions require agents to engage in both long-range prediction and interventional reasoning within the specified scene.
- **In-situ interventions**: The essence of interaction in physics dynamics lies in timing. Agents that deviate even slightly in timing at any step may potentially fail to complete the task.

`I-PHYRE` is a block elimination task; see Fig. 1 for illustrations. Agents must ensure all red balls fall into the hole by removing the minimum number of blocks possible. The benchmark incorporates varying levels of difficulty and is segmented into different generalization splits. We have formulated 40 distinct games and segregated them into four splits, including a basic split for training and three additional generalization splits. These are intended to assess the authentic interactive physical understanding of agents beyond mere data fitting. Specifically, the generalization splits are designed to test the agents' ability to (i) discern key physical elements amidst noise, (ii) strategize for long sequences through compositionality, and (iii) conform to more stringent timing constraints.

Importantly, agents must discern *when* to remove *which* block as the game progresses. In certain intricate scenarios, the sequence and timing of interventions become pivotal due to the perpetually evolving dynamics. To illustrate, we refer to Fig. 1 to analyze how interventions, varying in order and timing, can precipitate entirely disparate physical occurrences.

**Basic game**  Initially removing the block either to the right of or beneath the red ball will set it free; however, the latter approach will merely cause the ball to descend vertically, bypassing the hole, as depicted in *Failure: Wrong order*. The former strategy propels the ball downward to the right, necessitating agents to act promptly to eliminate the block below, directing the ball straight into the hole; this is illustrated in *Perfect solution: Use 2 steps*. An alternate tactic involves removing the two blocks on the right platform post the elimination of the top block, resulting in three steps; this scenario is portrayed in *Suboptimal solution: Use 3 steps*.

**Noisy game**  The introduction of an extraneous block should not alter the optimal strategy. The newly added gray block beneath the inclined block can be conveniently disregarded if the timing is accurately computed; this is illustrated in *Suboptimal solution: Disturbed by the noisy block*.

**Compositional game**  Compositional games amalgamate concepts found in basic games (e.g., events in *Compositional solution: Long-sequence planning*), leading to scenarios with an increased number of blocks and necessitating longer-range planning.

**Multi-ball game**  Introducing an additional ball to the scene elevates the level of challenge. The optimal solution requires an agent to simultaneously drop two balls on the long block and remove the block at the right moment, directing both balls into the hole; refer to *Perfect solution: Use 3 steps*. The crucial aspect here is the sequence in which the blocks on either side are removed; if the order of actions is inverted, the right ball will arrive at the block too late and will descend directly to the left platform, as seen in *Failure: Wrong order leads to missing elimination timing for two balls*. Nonetheless, one can still succeed in the game by swiftly adapting to the alteration and removing the top right inclined block at the correct moment; this is depicted in *In-situ adaptation*.

We conduct experiments utilizing three distinct planning strategies within I-PHYRE. (i) **Planning in advance**: the agent predefines a sequence of interventions at specific time steps, relying solely on the initial scene. (ii) **Planning on-the-fly**: the agent decides its current action based on the present scenario. (iii) **Combined strategy**: the agent recalibrates its planned timing after executing the earliest action. Both **supervised** and **reinforcement** agents undergo evaluation on I-PHYRE using a zero-shot generalization approach, being trained on the basic split and tested on the remaining three splits. To gain deeper insights into the capabilities of existing algorithms in interactive physical reasoning, we also establish a human baseline. The comparative analysis reveals that current learning algorithms are yet to match human proficiency in interactive physical reasoning, particularly regarding generalization (Grenander, 1993; Fodor et al., 1988; Lake et al., 2015; Zhu et al., 2007; George et al., 2017; Lake & Baroni, 2018).

In conclusion, our work renders three significant contributions:

- We delineate the challenge of interactive physical reasoning and unveil I-PHYRE, the first benchmark developed to scrutinize interactivity in physical reasoning.
- We devise three planning strategies for interactive physical reasoning (*i.e.*, planning in advance, planning on-the-fly, and combined) and implement with both supervised and reinforcement agents.
- We facilitate a comparative analysis between human performance and various learning algorithms, elucidating the challenges and charting the prospective avenues in interactive physical reasoning.

## 2 RELATED WORK

**Intuitive physics**  Research in intuitive physics is broadly bifurcated into two methodological streams. Traditional methods are anchored in the **violation-of-expectation (VoE)** paradigm (Spelke et al., 1992), where developmental studies gauge infants' intuitive physics acquisition by presenting them with events that either adhere to or violate physical laws, quantifying surprise through looking time at both event types (ee Baillargeon, 1987; Hespos & Baillargeon, 2001); a longer gaze typically indicates a perceived violation of physical laws. To instill comparable human-like physical intuition in learning algorithms, contemporary computational methods utilize simulators to replicate similar stimuli, evaluating whether modern learning models exhibit "surprise" at events that contravene

Table 1: **Comparison between `I-PHYRE` and related benchmarks.** `I-PHYRE` employs intuitive physics (instead of precise computation), showcases intricate dynamics, necessitates multi-step planning, demands accurate sequencing of interventions (*i.e.*, action order), and enforces impeccable timing of interventions *i.e.*, action timing) to resolve intricate problems.

| Benchmarks | mechanism | rich dynamics | multi-step | action order | action timing |
|---|---|---|---|---|---|
| Block Towers (Lerer et al., 2016) | intuition | ✗ | ✗ | ✗ | ✗ |
| ComPhy (Chen et al., 2022) | intuition | ✗ | ✗ | ✗ | ✗ |
| PHYRE (Bakhtin et al., 2019) | intuition | ✓ | ✗ | ✗ | ✗ |
| Virtual Tools (Allen et al., 2020) | intuition | ✓ | ✗ | ✗ | ✗ |
| SMP (Toussaint et al., 2018) | computation | ✗ | ✓ | ✓ | ✓ |
| `I-PHYRE` (ours) | intuition | ✓ | ✓ | ✓ | ✓ |

physical laws (Piloto et al., 2022). Importantly, these prediction-centric methods only emphasize **passive observation**, yielding only a rudimentary understanding of physical reasoning capabilities due to the absence of scene interactivity. The exploration of non-interactive physical reasoning also extends to trajectory prediction and question-answering (Yi et al., 2020; Chen et al., 2022).

Contemporary approaches address intuitive physics primarily through the perspective of **tool use** (Zhu et al., 2015; Toussaint et al., 2018), exemplified by frameworks like PHYRE (Bakhtin et al., 2019) and Virtual Tool Game (Allen et al., 2020). Even though tool use inherently necessitates multi-step planning (Toussaint et al., 2018; Zhang et al., 2022), prevailing benchmarks (Bakhtin et al., 2019; Allen et al., 2020) allow only a **single** interaction step with the scene. Consequently, computational models addressing these benchmarks predominantly develop supervised models, utilizing classifiers to distinguish between the success and failure of an action (Li et al., 2022) or incorporating dynamic prediction modules to inform judgments (Girdhar et al., 2020; Qi et al., 2021). However, none of the existing works explicitly incorporate any degree of interaction.

In summary, existing studies fail to thoroughly investigate the **interactive** aspect of physical reasoning in both humans and machines. This absence of interactivity hinders exploring **multi-step** physical reasoning. To bridge this gap, we introduce `I-PHYRE`, a new benchmark for interactive physical reasoning. Tab. 1 provides a comprehensive comparison between `I-PHYRE` and preceding works.

**Multi-step planning** Planning in intricate environments is a distinctive feature of intelligence. We direct readers seeking a broader understanding of this extensive field to a monograph on planning (LaValle, 2006) and a contemporary review (Garrett et al., 2021), while we narrow our focus to recent developments in multi-step planning that incorporate explicit modeling of physical constraints.

Sequential Manipulation Planning (SMP) stands out as a pivotal domain within multi-step planning. Contemporary research in this area addresses multi-step physical reasoning through (i) optimization in robotic simulators (Toussaint et al., 2018; 2020), (ii) adaptive sampling with tangible robots (Wang et al., 2021; Konidaris et al., 2018), or (iii) graph-based planning with visually grounded symbols (Jiao et al., 2022; Han et al., 2022). Although advancements have been made in multi-step planning, interaction with dynamic environments remains a formidable challenge; precise timing of interventions necessitates swift responsiveness as opposed to prolonged computation.

Contrastingly, `I-PHYRE` emphasizes swift, intuitive multi-step planning. The intricate dynamics (*e.g.*, falling, rotation, collision, friction, pendulum, elastic motion) and varied interactivity within `I-PHYRE` necessitate meticulous sequencing and impeccable timing of interventions—areas that have remained largely unexplored in the realm of physical reasoning.

**RL agents** Reinforcement Learning (RL) agents have achieved significant triumphs in the realm of gaming, surpassing human performance in strategic games post extensive training: (i) mastering Atari games (Mnih et al., 2015), (ii) conquering Go (Silver et al., 2016), and (iii) excelling in StarCraft (Vinyals et al., 2019). Despite the increasing complexity in action strategies (*i.e.*, evolving from Go and Atari to StarCraft), few environments explore fundamental physical reasoning beyond basic rules in games like Breakout. While some benchmarks incorporate physics, the emphasis predominantly lies on the generalization of actions (Jain et al., 2020). To counter this limitation, we introduce `I-PHYRE`, a novel challenge centered around physical reasoning, with a focus on the generalization of interactivity. During our experiments, we assess classic RL algorithms against `I-PHYRE`, revealing their limitations. A comparative study between humans and RL agents fosters discussions regarding prospective developmental trajectories in this domain.

## 3 CREATING I-PHYRE

### 3.1 GAME DESIGN

`I-PHYRE` encompasses 40 distinctive interactive physics games, each primarily featuring gray, black, and blue blocks, red balls, rigid sticks, and springs. All games share a unified objective: to guide all red balls into the hole by strategically eliminating gray blocks at different time steps, the only blocks that can be removed. Beyond the initial stationary blocks (gray and black), the games also incorporate movable blocks (blue) that react to gravity and impact, springs that can expand and compress, and rigid sticks that can roll within the environment. These elements collectively represent fundamental Newtonian mechanics, thereby enriching the physical dynamics of the games.

**Splits** Games are categorized into four splits—basic, noisy, compositional, and multi-ball—based on algorithmic stability and varying generalization levels (Xie et al., 2021). These splits are designed to assess abilities to (i) learn fundamental physical concepts; (ii) understand intuitive physics amidst noise; (iii) compose learned knowledge for long-sequence planning; and (iv) generalize across varying object quantities. For further details on game division and scene designs, refer to Appx. A. Agents are trained on the basic split and evaluated on the remaining generalization splits. Unlike PHYRE, which emphasizes data-driven approaches via extensive stochastic game data, our approach fosters robust generalization in physical reasoning through interaction with a select set of scenarios.

- **Basic split.** This split focuses on foundational physical concepts like angle, direction, fill, hinder, hole, impulse, pendulum, seesaw, spring, and support, essential for intuitive physics understanding and foundational for more intricate physical dynamics.

- **Noisy split.** Created to assess agents' resilience in physical reasoning against minor perturbations, this split tests the ability to maintain invariance to noisy blocks, indicative of a genuine understanding of physical dynamics. It includes games from the basic split, augmented with additional distractive gray blocks.

- **Compositional split.** Evaluating compositional generalization (Fodor et al., 1988; Lake & Baroni, 2018), this split combines known knowledge to solve novel tasks necessitating long-range reasoning. It contains games formed by chaining two basic split games or by merging two high-level concepts from the basic split to craft a new one.

- **Multi-ball split.** This split examines the agent's capability to manage multiple dynamic events concurrently, involving at least two balls. Success is achieved only when all balls are dropped into the hole, with dynamics being notably more intricate due to additional interactions among the balls, and action timings in successful solutions being more critical.

**Reward** To encourage learning efficient policies, we assign a negative reward of -1 per second and -10 for each gray block eliminated. Achieving the goal yields a substantial reward of 1000. The final score is the sum of accumulated rewards, promoting strategies that are both swift and action-efficient.

**Implementation** `I-PHYRE` games are implemented using pymunk, rendered in pygame at a $600 \times 600$ resolution, and integrated into Gym to streamline testing with advanced RL frameworks. The simulator conveys essential game information, including object positions, sizes, and pertinent indicators. For additional details regarding observation and action space, refer to Appx. C.

### 3.2 PLANNING STRATEGIES

Drawing inspiration from human proficiency in navigating games requiring intricate physical reasoning, we explore two planning strategies and their combination for formulating interactive physical reasoning problems: planning in advance, planning on-the-fly, and a combined strategy. These are realized within the frameworks of supervised learning and RL.

**Planning in advance** While `I-PHYRE` is fundamentally multi-step, it can be simplified through a one-step planning strategy. Here, agents observe the initial stationary scene and predict the timing to eliminate each block. Once initiated, agents adhere to this plan, executing block removals at predetermined moments. Practically, we employ deterministic agents, with their action space aligning with the timings at which blocks are eliminated.

**Planning on-the-fly**   Agents employing on-the-fly planning continuously interact with the environment, deciding actions at each step. This approach aligns with the classic RL framework, where agents, at each time step, propose a distribution of actions and select the one maximizing total rewards. By considering each observation as the state space, the blocks to be eliminated as the action space, and the physics simulator as the state transition function, the problem can be modeled as a standard Markov decision process.

**Combined strategy**   Agents can leverage a hybrid of the aforementioned strategies, benefiting from the relative reward density in planning in advance and the adaptability inherent in on-the-fly planning. In this strategy, agents, upon receiving the initial scene, determine the execution time for all actions, wait and execute the earliest action, and update the execution time based on the altered scene until the game's conclusion. The action space is also the timings at which blocks are eliminated.

## 4   EXPERIMENTS

Through a series of experiments, we assess the capability of current learning agents to acquire interactive behaviors and adapt to novel situations. We also present human performance as a benchmark for comparison. Additionally, we delve into the distinctions between various planning strategies within the context of interactive physical reasoning.

### 4.1   HUMAN BASELINE

**Procedure**   With an approved IRB, we recruited a total of 46 participants from a designated participant pool. Each participant played all 40 interactive physics games, aiming to achieve the highest scores. Prior to the experiments, participants were briefed that:

- The objective is to guide the red balls into the hole by removing gray blocks;
- Gray and black blocks are stationary, while blue blocks are movable due to gravity or impact;
- Scores are determined by the number of interventions and total time—fewer actions and quicker completion yield higher scores;
- Each game can be attempted a maximum of 5 times;
- Each game, once started, has a 15-second time limit, but pre-start contemplation time is unlimited;
- Failure is declared if the objective is not met within the time constraint.

The highest score achieved by each participant was automatically logged. The average score for each game was computed from the mean of participants' highest scores. To set an upper bound on these intricate tasks, scores achieved by the experimenters were considered the maximum attainable.

**Results**   Tab. 2 outlines the participants' performance relative to the oracle. The participants exhibit a success rate above 80%, demonstrating a robust ability to solve the majority of the games by effectively reasoning about in-

Table 2: **Rewards and success rates (SR) of participants and oracle.** Each player have five attempts per game.

| Player | Metric | Basic | Noisy | Compositional | Multi-ball |
|--------|--------|-------|-------|---------------|------------|
| Participants | Reward | 898.01 | 879.23 | 798.69 | 800.97 |
|  | SR (%) | 92.39 | 90.00 | 82.83 | 82.83 |
| Oracle | Reward | 976.06 | 974.71 | 971.26 | 968.61 |
|  | SR (%) | 100 | 100 | 100 | 100 |

teractive physical dynamics to attain specific goals. However, it is noteworthy that participants often do not achieve solutions that are optimal compared to the oracle, as quantitatively depicted in Tab. 2. A more detailed analysis of the results, available in Appx. J, indicates that participants find it challenging to solve certain compositional and noisy games with intricate dynamics, such as activated-pendulum and noisy-angle. Additionally, participants encounter difficulties in accurately timing interventions under stringent temporal constraints, as seen in games like impulse-spring and multi-ball-hinder.

### 4.2   LEARNING AGENTS

We employ the contemporary RL framework to model the dynamic decision-making. Specifically, we explore five model-free deep RL methods: PPO (Schulman et al., 2017), A2C (Mnih et al., 2016), SAC (Haarnoja et al., 2018), DDPG (Lillicrap et al., 2016), and DQN (Mnih et al., 2015). Additionally, we experiment with model-based RL, offline RL, and supervised learners to establish algorithm baselines. Refer to Appxs. D and E for their results.

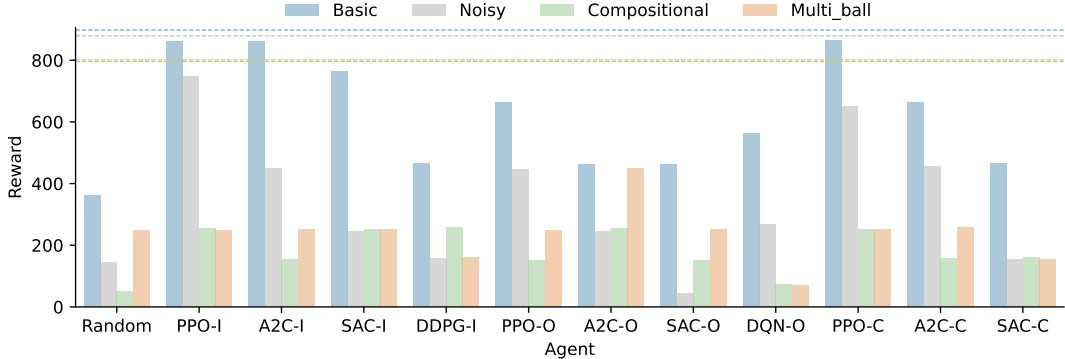

Figure 2: **Performance of various RL agents on `I-PHYRE`.** Agents, trained on the basic split, are evaluated on the remaining three splits in a zero-shot fashion. The suffixes '-I', '-O', and '-C' denote planning in advance, on-the-fly planning, and the combined strategy, respectively. The dashed lines are human results.

**Architectures** We configure PPO-I, A2C-I, SAC-I, and DDPG-I to follow the **planning in advance** paradigm (indicated by the suffix -I), as they are all apt for continuous action spaces, representing the timing of each action. Here, they predict the entire sequence of action timings once. During testing, the game engine takes in the action timing sequence, executing each action at the designated time.

Conversely, PPO-O, A2C-O, SAC-O, and DQN-O are structured as **planning on-the-fly** agents (indicated by the suffix -O), interacting with the environment in discrete time steps. Given the current state and action space, DQN-O selects an action from the action space, based on the action value determined by a neural network. The other three, being actor-critic-based agents, follow the same training pipeline as in planning in advance but generate a single action at each time step. This cycle continues until the goal is achieved or a terminal state is reached.

We also devise **a combined strategy** to mimic human planning. Specifically, agents, initially create a sequence of action timings based on the initial scene, similar to planning in advance, but adjust this action policy post-execution of the earliest action. For this approach, we utilize PPO-C, A2C-C, and SAC-C algorithms (indicated by the suffix -C).

### 4.2.1 HOW DO AGENTS PERFORM ZERO-SHOT GENERALIZATION ON DIFFERENT SPLITS?

Our assessment is centered on generalization to novel scenarios, scrutinizing agents' capabilities on noisy, compositional, and multi-ball splits after their training on the basic split, with no further fine-tuning. The generalization to similar tasks is trivial for agents as shown in Appx. F. The results, illustrated in Fig. 2, indicate that compositional and multi-ball splits are notably more demanding. They necessitate the ability to chain existing solutions effectively and to manage multiple physical objects with precise action timing for successful task completion.

Current RL agents manifest substantial gaps in generalization compared to humans. Humans display consistent and harmonious performance across all splits, highlighting a balanced development of interactivity-related capabilities, as detailed in Tab. 2. Conversely, some agents, such as DDPG-I, struggle to outperform even random agents. The data reveals a pronounced proficiency of RL agents in the noisy split, where their performance shows a correlation with their achievements in the basic split. However, this correlation diminishes in the compositional and multi-ball splits, where the inherent complexities of these tasks impact performance negatively. See Appx. B for analysis.

Distinct performance patterns are observed between humans and artificial agents across generalization splits. Humans maintain uniform performance, reflecting well-coordinated development of interactivity-related abilities. In contrast, artificial agents excel mainly in the noisy split, with no clear correlation with human performance, possibly due to task-specific variations and inherent complexities. The variability in the compositional and multi-ball splits underscores RL agents' unmet capabilities in interactive physical reasoning, as detailed in Appx. J.

### 4.2.2 HOW DO PLANNING STRATEGIES DIFFER?

Distinct characteristics of the three planning strategies significantly impact agent performance and generalization. As depicted in Fig. 3, agents using planning in advance converge more swiftly,

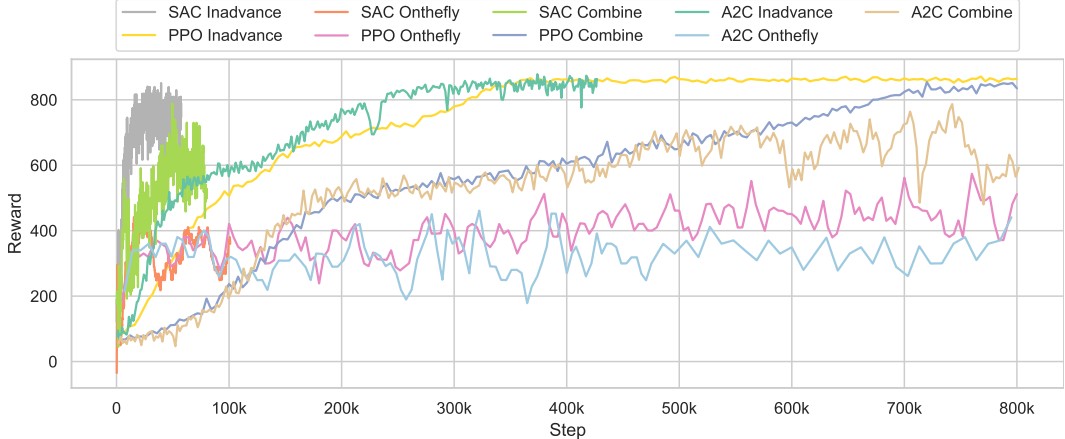

Figure 3: **The training curves of different RL agents on the basic split.** Training steps vary among agents.

stably, and effectively compared to on-the-fly planners, regardless of the specific agents used. This is likely due to the concise representation of action space, although it limits real-time decision-making capabilities.

In contrast, the oscillations in the training curves of on-the-fly planners may stem from the difficulties in learning sparse action distribution over prolonged periods, marked by few eliminations amidst numerous no-ops. The combined strategy, while converging slower than planning in advance, eventually reaches similar reward levels.

A closer look at Fig. 2 reveals that all strategies generalize similarly across different splits, with variations mainly in the basic split. The combined strategy, blending the temporal dimensions of planning in advance with updating mechanisms, offers enhanced adaptability and comparable generalization, even for planning on-the-fly agents, which, despite their learning challenges, emerge as viable contenders for interactive physical reasoning tasks.

## 5 DISCUSSION

`I-PHYRE` is designed to assess and advance the development of interactive agents capable of engaging in precise physical interactions through reasoning, planning, and intervention. We introduce three planning strategies and assess various classic learning agents, discussing challenges, limitations, and potential future directions below.

### 5.1 WHY CURRENT RL AGENTS FAIL ON `I-PHYRE`?

The findings in Sec. 4.2 reveal a substantial disparity between RL agents and humans in performance on `I-PHYRE`. We explore the potential reasons for this gap, considering both environmental and model design perspectives.

- **Physics modeling:** `I-PHYRE` incorporates extensive physical rules and concepts. However, existing RL agents, primarily focusing on mapping states to actions, lack an understanding of the objects' physical properties and the inherent physical commonsense within the scenes. Without comprehensively exploring all possible physical objects and state transitions, achieving a profound understanding of the governing physical rules remains elusive. The field has seen limited exploration into abstracting physical knowledge in RL agents, and even state-of-the-art methods employing GNN struggle with significant errors in unseen scenarios when predicting future trajectories under physical rules. We posit that physics modeling is pivotal for enhancing agent interaction with physical environments. More details are provided in Appx. I.

- **Multi-step interventions:** Contrasting the one-step nature of benchmarks like PHYRE (Bakhtin et al., 2019), `I-PHYRE` necessitates multi-step interventions, introducing longer delays in receiving feedback. This demands agents to learn patient interventions in the physical dynamics progressively. The delayed feedback may cause agents to struggle in discerning the efficacy of intermediate interventions in solving the `I-PHYRE` problems.

- **Action timing:** The exact timing requirements of `I-PHYRE` are challenging for RL agents, necessitating the identification and timely elimination of crucial blocks to optimize scores. The predominance of no-ops in action distribution makes current on-the-fly planning methods for RL agents overly cautious, and the environment's sensitivity to precise action timing introduces additional hurdles that are challenging for current agents to surmount. Further analysis of failure sources from action order and timing is discussed in Appx. H.

## 5.2 LEARNING TO JUDGE, PREDICT, AND ACT WITH OFFLINE DATA

The intuitive interaction with the physical environment can manifest through various learning forms. Beyond the realm of online RL, we delve into several offline learning tasks, allowing agents to leverage a repository of pre-collected data to make informed judgments, predictions, and actions.

Specifically, we scrutinize three learning paradigms: supervised learning, model-based RL learning, and offline RL learning, all operating as offline learners.

- **Supervised learners:** Functioning as offline learned discriminators, these learners assess whether a sequence of actions can resolve a task. However, the outcomes, detailed in Appx. D, depict their struggle to navigate new game splits without supplementary data.
- **Model-based learners:** These are offline learned predictors formulating projections of future states. Their efficacy is gauged by the rewards secured through the amalgamation of current and anticipated future states. Yet, as revealed in Appx. E, they don't exhibit notable advancements over their model-free RL counterparts.
- **Offline RL learners:** Acting as offline learned actors, these learners dictate actions based on the present states. However, the findings in Appx. E illustrate their inability to assimilate interactive principles from the available data.

In conclusion, our empirical investigations underscore the suboptimal performance of agents in offline learning scenarios, highlighting the pivotal role of real-time environments and adaptive exploration in learning interactive physical reasoning with current algorithms. Nonetheless, we conjecture that agents, when endowed with extensive physics modeling, can potentially excel in offline learning scenarios, enabling them to adeptly judge, predict, and act.

## 5.3 LIMITATIONS AND FUTURE WORK

We now discuss the limitations of our current approach and explore potential avenues for future research aimed at developing enhanced agents capable of interactive physical reasoning.

- **Optimal planning strategy:** While we have explored a combined strategy incorporating planning in advance and planning on-the-fly, it may not represent the pinnacle of human planning strategies. Future research could delve into optimizing the amalgamation of these strategies to model and augment advanced reasoning capabilities effectively.
- **Integration with large pre-trained models:** The rising interest in large pre-trained models within the community is noteworthy. Preliminary studies are conducted in Appx. G, and the fusion of these extensive pre-trained language models with quantitative methodologies could potentially enhance performance on `I-PHYRE`.
- **3D interactive environments:** Our study is centered around interactivity in a 2D physical environment to emphasize reasoning. However, developing a more realistic and interactive benchmark in 3D space could further propel the advancement of agents exhibiting human-like physical reasoning.

## 6 CONCLUSION

We present Interactive PHysical Reasoning (`I-PHYRE`) to assess learning methods in interacting with the physical world, focusing on intricate dynamics and precise action timing. We introduce three planning strategies and reveal a significant disparity between human capabilities and learning agents through extensive studies and experiments. The limitations and potential future directions are highlighted, with the hope that `I-PHYRE` will inspire advancements in understanding the interactive aspects of physical environments.

**Acknowledgment**   The authors would like to thank NVIDIA for their generous support of GPUs and hardware. This work is supported in part by the National Science and Technology Major Project (2022ZD0114900), the NSFC (62376009), and the Beijing Nova Program.

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

## A GAMES

We present the initial scenes of all the 40 games in Figs. A2 to A5. The games vary in different positional setups. We keep invariant the dynamic properties like density and friction to emphasize the interactive aspects of the task.

## B GAME DIFFICULTY IN TERMS OF SAMPLING

To evaluate the game difficulty in I-PHYRE, we measure the number of iterations required to gather 50 successful action sequences that do not repeat for each game. As an approximate estimation, the more iterations random sampling needs, the more difficult it is to solve; see Fig. A1 for results. The compositional games show significant difficulty compared with those in the other three splits, especially the seesaw angle and activated pendulum. These games have stringent requirement on execution timing: Players and agents may miss the perfect timing easily. Another interesting discovery is that the games related to the seesaw show higher difficulty regardless of the split. The analysis of game difficulty demonstrates a clear alignment with human performance, as humans tend to achieve lower scores in games that are more challenging to sample. See Tab. A4 for human scores in detail.

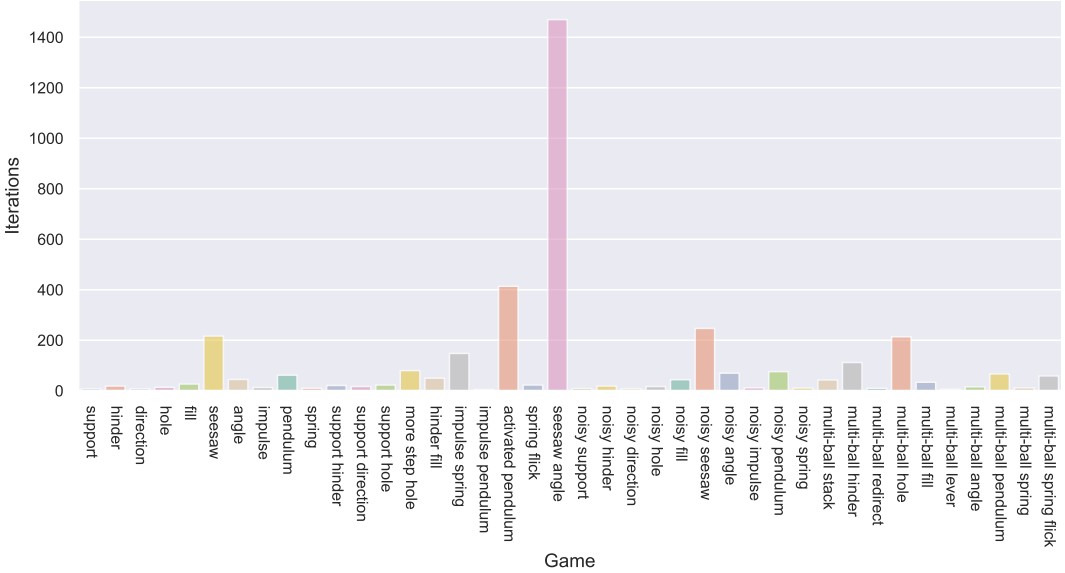

Figure A1: **The iteration numbers required to generate 50 different successful action sequences.**

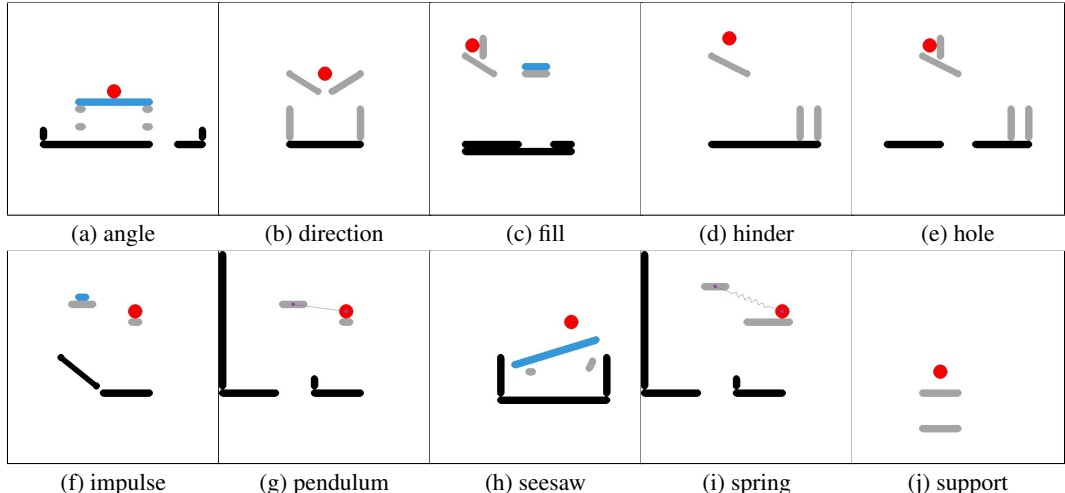

|            (a) angle | (b) direction | (c) fill | (d) hinder | (e) hole |

|            (f) impulse | (g) pendulum | (h) seesaw | (i) spring | (j) support |

Figure A2: **The initial scenes of basic games.**

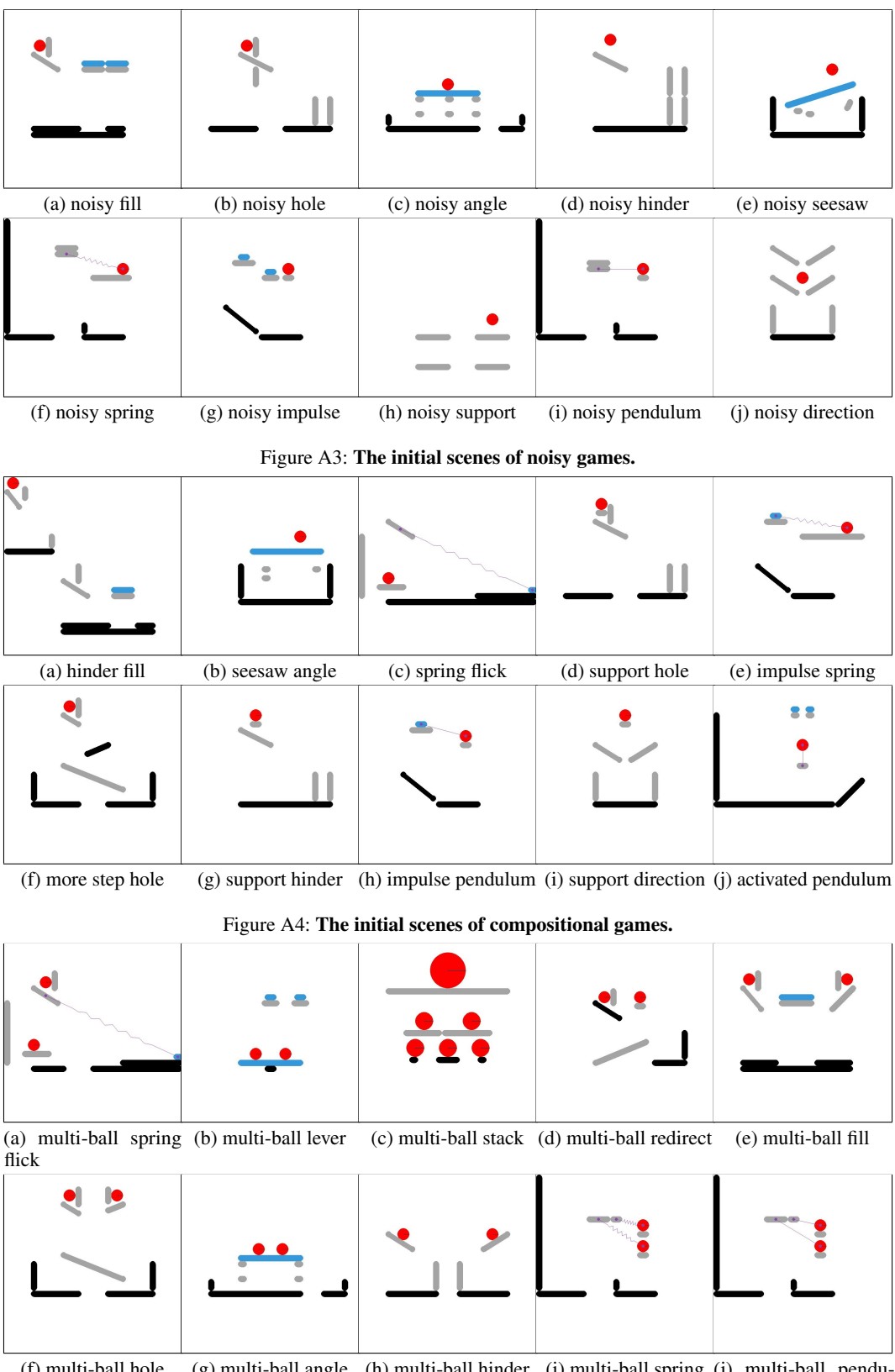

(a) noisy fill    (b) noisy hole    (c) noisy angle    (d) noisy hinder    (e) noisy seesaw

(f) noisy spring    (g) noisy impulse    (h) noisy support    (i) noisy pendulum    (j) noisy direction

Figure A3: **The initial scenes of noisy games.**

(a) hinder fill    (b) seesaw angle    (c) spring flick    (d) support hole    (e) impulse spring

(f) more step hole    (g) support hinder    (h) impulse pendulum    (i) support direction    (j) activated pendulum

Figure A4: **The initial scenes of compositional games.**

(a) multi-ball spring flick    (b) multi-ball lever    (c) multi-ball stack    (d) multi-ball redirect    (e) multi-ball fill

(f) multi-ball hole    (g) multi-ball angle    (h) multi-ball hinder    (i) multi-ball spring    (j) multi-ball pendulum

Figure A5: **The initial scenes of multi-ball games.**

## C  TRAINING DETAILS OF MODEL-FREE LEARNERS

We run all our experiments on RTX 3090 GPUs. The simulator produces the next state, reward, and termination indicator from the action per time step.

**Architectures**  We adopt three distinct strategies in training: planning in advance, planning on-the-fly, and the combined strategy.

In the planning in advance strategy, the combination of initial observation of the game and the entire action space serves as the model's input, while the output consists of a continuous-valued vector whose dimension equals to the maximum number of actions (6 in our case). Each vector element represents the normalized action timing. We implement model-free reinforcement learning algorithms, namely Proximal Policy Optimization (PPO-I), Advantage Actor-Critic (A2C-I), Soft Actor-Critic (SAC-I), and Deep Deterministic Policy Gradient (DDPG-I), to generate continuous action values in accordance with this setup.

In the planning on-the-fly strategy, at each time step, the model's input comprises the current observation combined with the entire action space, while the output consists of the probability for each possible action, including no action. The action with the highest probability is executed during inference time at each step. Following this approach, we implemented model-free reinforcement learning algorithms Proximal Policy Optimization (PPO-O), Advantage Actor-Critic (A2C-O), Soft Actor-Critic (SAC-O), and Deep Q-Network (DQN-O).

For the combined strategy, the model's input is the fusion of the current game observation and the entire action space, while the output is the same as that of the planning in advance strategy. This single-step procedure is analogous to the planning in advance strategy; however, after executing the first action, the entire action distribution is updated based on the current observation. Employing this framework, we implemented model-free reinforcement learning algorithms Proximal Policy Optimization (PPO-C), Advantage Actor-Critic (A2C-C), and Soft Actor-Critic (SAC-C).

The policy architecture of model-free learners is MLP with two hidden layers of size 256 and the activation function is tanh.

**Learning**  The observation for a scene is processed in the symbolic space, represented as a $12 \times 9$ matrix. Each row denotes one object's features in the scene. We use the symbolic representation of objects instead of visual input since symbolic representation consists of all the necessary components to do reasoning and visual information does not contribute any additional useful information for this particular task but introduces uncertainty and noise into the planning process.

The object feature vector consists of all the essential components for interactive physical reasoning:

- **Object Position**: The spatial coordinates of the objects with four scalars representing the two endpoints for bars or two duplicate centers for balls.
- **Object Size**: The radius of the object.
- **Eliminable Indicator**: An indication of whether the object can be eliminated in the given context.
- **Fixed Object Indicator**: The identification of whether the object is stationary and can not be moved due to gravity.
- **Joint Indicator**: An indicator of whether the object is connected to a joint.
- **Spring Indicator**: An indicator of whether the object is connected to a spring.

The action space is the concatenation of the positions of the objects that can be removed from the scene, padded to the maximum number of 6. Since different games have different action spaces, we concatenate the action space with the observation space as input to enable generalizable reasoning with a single agent.

The same object features and action space are used in the other learning paradigms as well.

A2C-I, A2C-C, and SAC-C are trained with a learning rate of $1 \times 10^{-5}$. All other models are trained with a learning rate of $1 \times 10^{-6}$. SAC-I is trained for 57k steps. SAC-O and SAC-C are trained for 80k steps. A2C-I are trained for 426k steps. Other models are trained for 800k steps.

**Results**  Please refer to Sec. 4.2 for analysis. The detailed rewards are listed in Appx. J.

## D    SUPERVISED LEARNERS

A commonly used metric in evaluating physical reasoning (Qi et al., 2021; Bakhtin et al., 2019; Girdhar et al., 2020; Li et al., 2022) is the accuracy of a classifier model to correctly predict the outcome of an action. In this part, we follow the same protocol: whether a supervised classifier can predict the outcome of an action sequence based on the initial scene.

**Architectures**    We formulate the problem as a binary classification task, wherein a model predicts whether an action sequence will succeed. We consider three different general architectures: Global Fusion, Object Fusion, and Vision Fusion. Of note, this paradigm falls into the category of planning in advance. Each model takes as input the action sequence and the initial scene configuration and outputs the success probability. The Global Fusion model embeds the entire action sequence and all object states and fuses them together using multiple MLPs. The Object Fusion model embeds each action and symbolized object independently and fuses them, both using MLPs. The Vision Fusion model is similar to the Global Fusion model except that it takes the pre-trained ViT (Dosovitskiy et al., 2020) features of the initial scenes as extra inputs.

**Learning**    We generate 50 successful and 50 failed action sequences for each game by random sampling. The generation iteration is regarded as an approximate estimate of the difficulty of games. To simplify the action space, we discretize 15 seconds into 150 time steps. Models are trained on the basic split and are tested on the other three splits. The object features are the same as the ones in Appx. C. The supervised agents are trained for 200 epochs with a batch size of 16, with the learning rate annealed from $1 \times 10^{-3}$ to $1 \times 10^{-6}$ using a cosine scheduler.

**Results**    Tab. A1 tabulates the performance of each supervised learning model, measured by mean accuracy across the test games. Of note, a random classifier should reach about 50% accuracy due to an equal number of positive and negative samples. However, all supervised learning agents fail to show notably improved performance compared with a random guess. Among the models, the Object Fusion model shows the best generalization compared to others due to fine-grain embeddings for actions and object states. Adding visual features from a pre-trained ViT does not necessarily improve the overall performance in the Vision Fusion model, though we do find enhanced accuracy when generalized to multiple balls.

Table A1: **Performance of different supervised learning models on `I-PHYRE` measured by mean accuracy (%).** Models are trained on the basic split.

| Agent | Bas. | Noi. | Comp. | Mul. |
|-------|------|------|-------|------|
| Global Fusion | 87.5 | 59.8 | 57.0 | 56.7 |
| Object Fusion | 71.4 | **60.1** | **60.2** | 54.1 |
| Vision Fusion | 86.4 | 57.5 | 55.8 | **59.5** |

These experimental results demonstrate that training supervised learning models on naively sampled data cannot endow agents with interactive physical reasoning ability.

## E    MODEL-BASED AND OFFLINE RL LEARNERS

We apply the model-based World Model (Ha & Schmidhuber, 2018) and the offline Decision Transformer (DT) (Chen et al., 2021) to the `I-PHYRE`.

**Architectures**    In the model-based method, we pre-train an MDRNN to predict the next state given the current state and action. The predicted states are concatenated with the current states as guidance for learning policies. We use PPO, SAC, and A2C as controllers. In the offline method, we use GPT-2 as the backbone to learn the mapping from states to actions autoregressively.

**Learning**    We used 50 successful and 50 failed actions per game to train the model-based and offline models. Models are trained on the basic split and tested on the other three splits. For the model-based learner, the prediction module of MDRNN is trained for 50 epochs with a learning rate of $1 \times 10^{-3}$, a batch size of 128, and a sequence length of 64. The prediction module is then fixed and served as an additional part of the observation. The offline RL learner is trained for 100 epochs with a batch size of 128, with the learning rate annealed from $1 \times 10^{-3}$ to $1 \times 10^{-6}$ using a cosine scheduler.

**Results**    As shown in Fig. A6, the results indicate that the model-based method doesn't improve the performance of PPO, SAC, and A2C, potentially due to the fact that RNN cannot learn an appropriate

physical dynamics model to accurately predict the next state. The offline method is also unsatisfactory; the Decision Transformer planning on-the-fly learns a conservative strategy that takes no action at all. Although the performance may improve by increasing the data size, we argue that delicate modeling of physics is essential to learning better dynamics and emerging interactive physical reasoning.

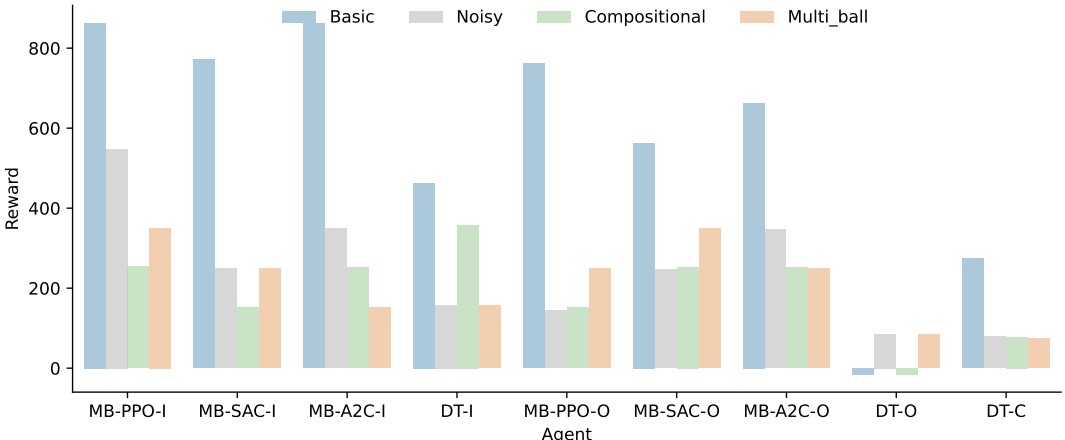

Figure A6: **Performance of model-based and offline RL learners.**

## F  WITHIN-TEMPLATE GENERALIZATION

To see the generalization capabilities of current intelligent agents in games with similar structures, we developed an unseen basic split consisting of an additional 10 games. These games resemble the basic ones but differ in object positions and angles. As with the previous setting, we evaluate A2C-I, PPO-I, A2C-C, PPO-C, A2C-O, and PPO-O that are trained on the basic split. The agents' performance in these modified games is detailed in Tab. A2. The performance nearly matches the basic split but exceeds the performance of the noisy, compositional, and multi-ball splits. These results indicate that current agents are adept at generalizing similar tasks but struggle with generalization across different game templates. Thus, we focus specifically on the challenge of generalizing to entirely new tasks in I-PHYRE.

Table A2: **Performance of agents on 10 similar tasks to basic games, measured by average rewards.**

| Split | Random | A2C-I | PPO-I | A2C-C | PPO-C | A2C-O | PPO-O |
|---|---|---|---|---|---|---|---|
| Basic | 360.33 | 862.47 | 861.56 | 765.26 | 862.78 | 461.2 | 663.91 |
| Unseen basic | 360.17 | 862.25 | 763.13 | 662.72 | 660.51 | 561.75 | 662.12 |

## G  LARGE LANGUAGE MODEL

In this interactive physical reasoning environment, GPT-4 is tasked not only to do physical reasoning but also to plan and take actions at specific times. This is quite a challenge for GPT-4 with only the initial states as input configuration. We prompt GPT-4 with detailed game rules and object features. The results show that GPT-4 can successfully finish some preliminary tasks like support and noisy support but fail on other tasks. The average rewards of different folds are in Tab. A3.

Table A3: **Performance of GPT-4 on I-PHYRE measured by average rewards.**

| Agent | Bas. | Noi. | Comp. | Mul. |
|---|---|---|---|---|
| GPT-4 | 75.17 | 64.17 | -29.10 | 77.49 |

The preliminary results suggest that the large language model cannot perform well in this interactive reasoning task, although they may be good at understanding physical concepts and utilizing physical heuristics. The interactive physical reasoning scenarios challenge GPT-4 to reason on both spatial information and precise action timing. Studies need to combine quantitative methods to further extend its capability to do physical reasoning, especially in terms of interactive reasoning.

## H ANALYSIS ON FAILURE CASES

To gain more insight into the interactivity in physical reasoning, we delve into failure cases specifically related to action order and timing. We assume that, to solve the game in an oracle way, one should first decide the correct action order and then the precise action timing. The situations when actions in the wrong order can still solve the game are beyond our discussion. Thus, the failure may come from two sources: (i) wrong action order and (ii) wrong action timing with the correct order. We want to explore to what extent the agents failed due to those two reasons respectively. We benchmark against established baselines: PPO, A2C, and SAC. We count three terms:

- **Success Number (SN):** The number of scenes in a split where agents could solve puzzles.
- **Right Order Number (RON):** The number of scenes in a split where agents aligned with the Oracle in terms of action order.
- **Success from Right Order Number (SRON):** The number of scenes that the agent solved and also matched the Oracle's action order.

With a total number of games $Total$, the percentage in failure cases of wrong action timing with the correct order, denoted as $P(T \mid F)$, can be calculated as:

$$P(T \mid F) = \frac{RON - SRON}{Total - SN}. \tag{A1}$$

The percentage in failure cases with wrong action order, denoted as $P(O \mid F)$, can be calculated as:

$$P(O \mid F) = 1 - P(T \mid F). \tag{A2}$$

The percentage of two failure sources in PPO, A2C, and SAC are shown in Fig. A7.

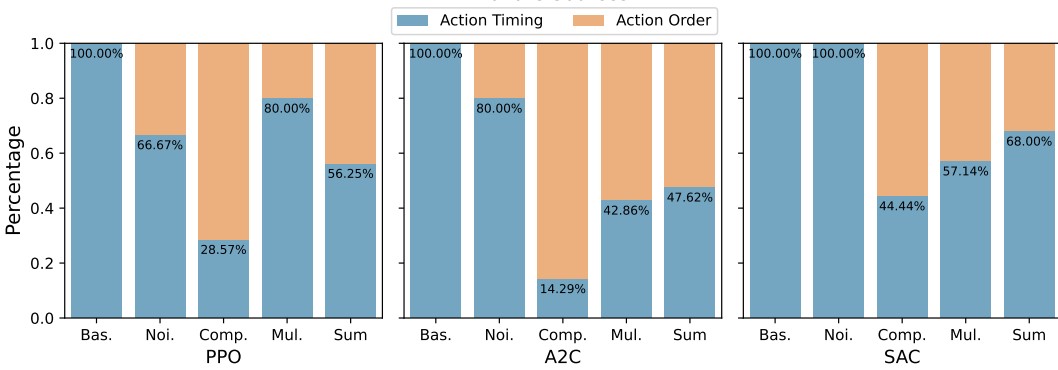

Figure A7: **The percentage of two failure sources in I-PHYRE.**

Our findings suggest that, of the cases solved, many of them are executed with the optimal action order. For cases not solved, about half of them are due to correct action order but wrong action timing on average (56.25%, 47.62%, and 68.00%), suggesting that both action order and action timing are important. When examining basic games, we find that learning action order is much easier than learning action timing since all of the failures come from wrong action timing. Additionally, for compositional games, the main challenge lies in executing actions in a more reasonable order, while in noisy games, the primary hurdle is enhancing the accuracy of action timing. In summary, our observations indicate that while action order significantly influences puzzle-solving success, the nuances of action timing prove particularly challenging for RL agents to get better performance. We hope this analysis can provide some of the future directions for model design from the temporal aspect.

## I DETAILED DISCUSSION ABOUT PHYSICS MODELING

Physics modeling serves as a crucial tool for comprehending and engaging with the physical realm, and its integration is critical in the progression of machine learning agents (Zhu et al., 2020; Duan

et al., 2022). The primary branches of physics modeling include physics-based simulation, physics-informed methodologies, and intuitive commonsense modeling.

The implementation of physics modeling can be achieved through physics-based simulations (Battaglia et al., 2013; Kubricht et al., 2016). Due to the recent developments in computational capabilities and efficiency, it is now possible to accurately simulate an array of physical phenomena in real time. These physics engines offer a consistent and comprehensive platform where agents can employ these principles to foresee future states. Despite its practicality, physics-based simulation presents its own challenges, especially regarding the computational demands for precisely depicting complex physical systems and grasping latent variables in partially observed environments (Ludwin-Peery et al., 2021). Moreover, it necessitates expert knowledge to construct, and it's not capable of learning from experience.

Another method is incorporating explicit physics constraints directly into the agent by employing physics-informed neural networks (Raissi et al., 2019; Cuomo et al., 2022). Physical laws and principles are used as constraints or guides in the learning process. It can help in building more robust models that are less prone to overfitting and ensure that the model's predictions adhere to established physical laws, making them more interpretable and reliable. This approach helps mitigate accumulated estimation errors. However, it might not generalize well to unseen scenarios due to its reliance on predefined physical laws.

Supplementing learning of physical dynamics with commonsense reasoning is another method. Agents learn not just how object states evolve, but also the commonsense or intuitive understanding of how these objects behave under certain circumstances (Piloto et al., 2022; Kubricht et al., 2017; Dasgupta et al., 2021; Weihs et al., 2022). This involves leveraging the vast amount of implicit knowledge humans possess about the world and encoding it into agents and could potentially lead to more robust and generalizable models.

However, each method has its challenges and limitations, and further research is required to develop more efficient and effective ways of incorporating physics modeling into agents.

## J COMPLETE RESULTS OF HUMANS AND RL AGENTS

We show the detailed game rewards from RL agents, human participants, and a random agent. For straightforward comparison, we average the rewards in 40 games; see Tab. A4 for details.

Table A4: **All results of humans and different RL agents on I-PHYRE benchmark, measured by rewards of gameplay.**

| Game | Human | Random | DDPG-I | DQN-O | A2C-I | A2C-O | A2C-C | PPO-I | PPO-O | PPO-C | SAC-I | SAC-O | SAC-C |
|---|---|---|---|---|---|---|---|---|---|---|---|---|---|
| support | 975.96 | 977.6 | 968 | 970.3 | 973.6 | 970.6 | 974.3 | 973.6 | 977.6 | 973 | 969.6 | 977.6 | 975.9 |
| hinder | 971.57 | -45 | -25 | 972.5 | 962.5 | 962.5 | 962.5 | 962.5 | 962.5 | 962.5 | 962.5 | -45 | 972.5 |
| direction | 971.89 | 953.3 | -55 | 947.3 | 958.4 | 963.3 | 959.7 | 949 | 953.3 | 948.9 | 963.3 | 953.3 | 970 |
| hole | 966.08 | -55 | 955.1 | 952.5 | 950.1 | -55 | -55 | 949.7 | 962.5 | 960 | 959.6 | 953.9 | 966 |
| fill | 970.89 | -45 | -35 | -45 | 957.1 | -45 | -35 | 958.4 | -45 | 958.6 | -45 | -45 | 969.5 |
| seesaw | 436.36 | -35 | -25 | -35 | -35 | -35 | -35 | -35 | -35 | -35 | -35 | -35 | -35 |
| angle | 770.47 | -55 | -45 | 948.2 | 952.9 | 949.3 | 962.9 | 950 | 949.4 | 950 | 958.3 | 951.6 | -45 |
| impulse | 970.47 | 972.3 | 966.5 | 973.5 | 967.5 | 971.3 | 967.8 | 967.8 | 972.3 | 967.7 | 966.9 | 972.1 | -35 |
| pendulum | 972.56 | -35 | 969 | -35 | 966.9 | -35 | 971.7 | 968.7 | -35 | 971.2 | 968.2 | -35 | -35 |
| spring | 973.84 | 970.1 | 969.6 | -35 | 970.7 | -35 | 970.6 | 970.9 | 976.5 | 970.9 | 984.2 | -35 | -35.1 |
| noisy_support | 977.09 | 957.6 | 949 | 947.9 | -45 | 957.6 | -45 | 952.5 | 954.4 | 953.8 | 952.9 | 957.6 | 960.3 |
| noisy_hinder | 967.33 | -65 | -55 | 952.5 | 951.9 | 942.3 | 962.5 | 938.9 | 942.5 | 952.5 | -65 | -65 | -45 |
| noisy_direction | 972.81 | 928.7 | -45 | -25 | 940.6 | -75 | 950.2 | 928.5 | -75 | 938.6 | 940.1 | -75 | -55 |
| noisy_hole | 966.43 | -65 | 955.4 | -55 | 950.3 | 946 | 959.5 | 941.3 | 942.5 | 951.5 | -65 | -65 | -65 |
| noisy_fill | 971.47 | -55 | -45 | -15 | -45 | -55 | -45 | -45 | -55 | -55 | -55 | -55 | -55.1 |
| noisy_seesaw | 455.88 | -45 | -25 | -45 | -45 | -45 | -45 | -45 | -45 | -45 | -45 | -45 | -35 |
| noisy_angle | 565.88 | -75 | -45 | -25 | -45 | -75 | 931.4 | 935.3 | -75 | 941.3 | -75 | -75 | -45 |
| noisy_impulse | 968.36 | -45 | -35 | -35 | 956.7 | -45 | 957 | 956.2 | 962.3 | 956.9 | -45 | -35 | -45 |
| noisy_pendulum | 972.59 | -45 | -35 | -15 | -45 | -45 | -45 | 961.4 | -45 | -45 | -45 | -45 | -45.1 |
| noisy_spring | 974.47 | -45 | -35 | 983.1 | 960.8 | -45 | -45 | 961.9 | 966.6 | 963.3 | 966.1 | -45 | 984.3 |
| support_hinder | 961.54 | -55 | -55 | -25 | 946.1 | -55 | -45 | 946.8 | -55 | 945.9 | -45 | -55 | -35 |
| support_direction | 963.07 | -65 | 956 | -45 | -35 | -65 | 951.3 | -55 | -65 | 936.2 | -65 | -65 | -55 |
| support_hole | 957.88 | -65 | -45 | -25 | -65 | 945 | -45 | -65 | -65 | -65 | -65 | -65 | -45.1 |
| more_step_hole | 969.51 | -45 | -45 | -25 | -45 | 965 | -45 | 973.5 | 963.6 | -45 | -45 | -45 | -45 |
| hinder_fill | 954.2 | -75 | -55 | -35 | -65 | -75 | -55 | -65 | -75 | -65 | 938.3 | -75 | -55 |
| impulse_spring | 281.96 | -35 | -35 | -25 | -35 | -35 | -35 | -35 | -35 | -35 | -35 | -35 | -35 |
| impulse_pendulum | 975.79 | 972.8 | 966.3 | -25 | 966.3 | -35 | 969.2 | 966.5 | 973.4 | 966.2 | 967.5 | 972.3 | 968.7 |
| activated_pendulum | 577.84 | -45 | -45 | -15 | -45 | -45 | -45 | -45 | -45 | -45 | -45 | -35 | -45 |
| spring_flick | 935.83 | -45 | 966 | 966 | -45 | 974.9 | -35 | -45 | -45 | -45 | 958.6 | 963.9 | 975.1 |
| seesaw_angle | 409.22 | -45 | -35 | -15 | -45 | -45 | -45 | -45 | -45 | -45 | -45 | -45 | -35 |
| multi_ball_stack | 612.59 | -45 | -35 | -25 | -35 | 969.9 | -35 | 958.5 | -45 | -45 | -45 | 957.3 | -45.1 |
| multi_ball_hinder | 487.67 | 948 | -45 | -25 | -45 | -45 | -55 | -55 | -55 | -55 | -55 | -55 | -55 |
| multi_ball_redirect | 919.63 | 964.7 | -35 | -15 | 961.1 | 962.4 | 961.2 | -45 | 964.5 | 962.4 | 955.5 | 964.5 | -35 |
| multi_ball_hole | 673.51 | -65 | -45 | -45 | -65 | -65 | -55 | -55 | -65 | -55 | -65 | -65 | -65 |
| multi_ball_fill | 957.84 | -65 | -45 | -45 | 936.8 | -65 | -55.1 | -65 | 936.1 | -45 | -65 | -65 | -65 |
| multi_ball_lever | 980.33 | 969.7 | 990.8 | 990.8 | 968.9 | 969.9 | 970.8 | 969.1 | 974.5 | 970.7 | 980.8 | 976.9 | 980.8 |
| multi_ball_angle | 927.3 | -55 | -45 | -45 | -55 | 951.5 | -45 | -55 | -55 | -55 | -55 | -55 | 969.7 |
| multi_ball_pendulum | 928.52 | -55 | -45 | -35 | -55 | -55 | -45 | -55 | -55 | -55 | -55 | -55 | -45 |
| multi_ball_spring | 896.18 | -55 | -45 | -25 | -45 | -55 | 960.5 | 947.9 | -55 | 950.9 | 960.6 | -55 | -45.1 |
| multi_ball_spring_flick | 624.17 | -55 | 956 | -25 | -55 | -55 | 945.9 | -45 | -55 | -55 | -45 | -55 | -55 |
| average reward | 844.17 | 200.87 | 260.19 | 242.99 | 429.36 | 352.69 | 383.45 | 528.10 | 377.74 | 504.33 | 378.45 | 227.15 | 233.93 |
| success rate | 87.55% | 25.00% | 30.00% | 27.50% | 47.50% | 40.00% | 42.50% | 57.50% | 42.50% | 55.00% | 42.50% | 27.50% | 27.50% |

