# OpenReview forum: "I-PHYRE: Interactive Physical Reasoning"
_ICLR.cc/2024/Conference — ICLR 2024 poster_

### Official Review · Reviewer_PPjJ · 2023-10-24

**Soundness:** 3 good
**Presentation:** 3 good
**Contribution:** 3 good
**Rating:** 6
**Confidence:** 4

**Summary:**

The paper studies the problem of interactive physical reasoning with focus on intuitive physical reasoning (approximate capability to predict physical outcome), multi-step planning (execution of multi-step actions to complete the task), in-situ interventions (necessity for timely object manipulation to succeed). To study this problem authors propose a block elimination task where the task is to ensure all red balls fall into the hole by removing minimum number of blocks. The benchmark consists of 40 unique games segmented into 4 splits: Basic split for training and noisy, compositional and multi-ball splits for testing generalization of physical understanding of agents. Authors also propose 3 different planning strategies: planning in advance, planning on the fly, and combined strategy to solve I-PHYRE using both supervised and reinforcement learned agents. In addition, authors also benchmark human performance on the task and compare it with current learning algorithms.

**Strengths:**

1. Proposed problem statement is interesting and relevant to advance physical understanding ability of learned agents. The proposed block elimination task captures the multi-step planning and necessity for timely actions which are novel aspects of the benchmark
2. The experimental setup demonstrates performance of 3 strong baselines using different learning paradigms i.e. reinforcement and supervised learning. It also demonstrates effectively that proposed baselines significantly underperform and there’s a lot of room for improvement
3. The paper benchmarks and establishes a human baseline for interactive physical reasoning
4. The paper is well written and easy to follow

**Weaknesses:**

1. It is unclear if the learned agents can generalize to unseen games from the basic split. The benchmark tests generalization to noise, compositionality and multi-ball setup but doesn’t present results on unseen games with properties similar to basic split. It would be good if the authors can add a unseen basic split and present performance of trained agents on unseen basic split. The Noisy split seems like a substitute of basic split but as it is using same games from basic split + some noise I am worried it is not a good representative of generalize to unseen games with similar properties
2. The training dataset of 20 games for basic split seems too small a dataset to lead to any meaningful generalization to complex splits on which generalization is being tested. Can authors describe why they chose a small dataset for training? Is it possible to procedurally scale the training data? If not, why?
3. Results presented in fig. 2 and 3 only show comparison of different methods on average reward on evaluation split it doesn’t highlight what is the success rate of these baselines on full task. It would be nice to have a comparison on achieved success on the full task to get a better sense of the results.
4. Poor performance on compositional and multi-ball splits for RL trained baselines look as expected with MLP policies and no large scale RL training. Can authors elaborate more on whether the I-PHYRE benchmark expects emergence of compositional generalization to such complex splits by training on just the basic split games? If yes, have authors tried using recurrent policies? Eventhough limited but recurrent networks have demonstrated some form of compositional generalization [1] in limited machine translation tasks which require “mix-and-match” strategies to solve the task (which is true for I-PHYRE compositional split)
5. The proposed task operates in a simple 2D environment which is less realistic setup

[1] Lake, B. M., & Baroni, M. (2018) Generalization without Systematicity: On the Compositional Skills of Sequence-to-Sequence Recurrent Networks

**Questions:**

1. How is the combined strategy baseline implemented and what is the action space of this baseline? From the text in main paper it seems like this baseline predicts continuous values at the start of the episode and keeps updating the full vector after each timestep. Is this correct?
2. Why are the combined strategy baseline trained only for < 100k step in Fig. 3 experiments?
3. Have authors tried using a on-the-fly version of GPT-4 baseline presented in the appendix? How well does that perform? By on-the-fly version, I mean querying GPT-4 after each timestep to output the next action instead of using just the initial scene

The primary concern I have is around the small size of training dataset and the challenges around scaling it. I'd appreciate if authors can discuss issues and concerns around that question. I am open to updating my rating if authors answer my questions.

---

> ### Author Response · Authors · 2023-11-17
> **Response to Reviewer PPjJ**
>
> We thank Reviewer PPjJ for constructive feedback. Below, we answer questions raised by PPjJ and point out several misunderstandings.
>
> > Generalize to unseen games from the basic split.
>
> We further created another 10 games similar to the basic games by varying the object positions and angles. Agent's generalization performance is shown in the following table. Overall, the performance in these new games is slightly inferior to the original basic games but better than noisy games.
>
> | **Agent**        | Random | A2C-I  | PPO-I  | SAC-I  | A2C-C  | PPO-C | SAC-C | A2C-O | PPO-O | SAC-O |
> |-|-|-|-|-|-|-|-|-|-|-|
> | **Reward**       | 360.17 | 862.25 | 763.13 | 462.48 | 662.72 | 660.51 | 363.60 | 561.75 | 662.12 | 461.39 |
>
> > The training dataset of 20 games for basic split seems too small a dataset to lead to any meaningful generalization to complex splits.
>
> We choose not to scale up I-PHYRE for the following reasons.
>
> 1. **Prior evidence does not support large-scale learning approach**: As experimental results from the original PHYRE show, existing methods that excel in within-template tasks still fall short in cross-template settings, despite the fact that the agent learns from a vast amount of variations/data.
> 2. **Minimal yet complete system for probing the boundary of intuitive physics models**: Indeed, one can scale up by introducing stochasticity into the game dynamics. However, naive scaling risks creating unsolvable games, given that the games demand precise timing. Critically, one of the central goals of this paper is to probe the boundary of the intuitive physics models. We argue the presented simulator is already a minimal yet complete system to achieve this goal.
> 3. **Learning generalizable and compositional physics**: Given the generalization results presented in works like PHYRE, we set our primary goal in I-PHYRE to be agents that can generalize to unseen scenarios from **learning physical primitives and composing them** with well-designed modeling rather than merely data-driven learning. This paradigm has been justified in other work, such as the [schema network (ICML 17')](https://arxiv.org/pdf/1706.04317.pdf), where the model learns from only a single setup and becomes generalizable to a wide range of variations.
>
> > The success rate of these baselines on full task.
>
> Thank you for pointing it out. We list the success rates of the methods below and will update this table in revision.
>
> | **Agent**        | Human   | Random | DDPG-I | DQN-O  | A2C-I  | A2C-O  | A2C-C  |
> |-|-|-|-|-|-|-|-|
> | **Success Rate** | 87.55% | 30.00% | 25.00% | 45.00% | 50.00% | 42.50% | 55.00% |
>
> | **Agent**        | PPO-I  | PPO-O  | PPO-C  | SAC-I | SAC-O | SAC-C |
> |-|-|-|-|--|--|-|
> | **Success Rate** | 57.50% | 47.50% | 55.00% | 37.50% | 40.00% | 37.50% |
>
> > Whether I-PHYRE expects emergence of compositional generalization by training on just the basic split games? If yes, have authors tried using recurrent policies?
>
> We do believe that compositional and systematic generalization can emerge by learning from basic games, as evidenced and justified by [schema network (ICML 17')](https://arxiv.org/pdf/1706.04317.pdf), [MLC (Nature 21')](https://www.nature.com/articles/s41586-023-06668-3), and [RCN (Science 17')](https://www.science.org/doi/10.1126/science.aag2612). Note that we do not commit to either MLPs or recurrent networks but rather believe in careful model design.
>
> To directly answer the reviewer's question, we indeed have tried recurrent networks to build a model-based RL agent. However, we observe no significant performance difference; see Appendix E. We conclude that the lack of physics modeling hinders learning helpful dynamics. We discuss potential ways of physics modeling in Appendix H. We will further discuss Lake and Baroni's work in revision.
>
> > How is the combined strategy baseline implemented and what is the action space of this baseline?
>
> The combined baseline predicts the execution time for all actions, waits and executes the earliest action, and updates the execution time. The action space per time step is the same as other strategies, which are all the blocks.
>
> > Why are the combined strategy baseline trained only for < 100k step in Fig. 3 experiments?
>
> The ones that have < 100k steps in Fig. 3 are SACs, including Inadvance, Combine, and Onthefly. This is because different RL algorithms are set with different numbers of **steps per iteration** implemented in Ray RLlib. SAC is set with 1 by default and A2C with 10 by default. However, the **number of iterations** remains the same across different algorithms.
>
> > Have authors tried using a on-the-fly version of GPT-4 baseline presented in the appendix?
>
> In our preliminary study, we tried basic games using GPT-4 with a planning on-the-fly strategy. The GPT-4 solves none of them and almost always returns no action. Oftentimes, it regards a new frame as no change from the previous one, as there are only limited changes in locations in between.

---

> ### Comment · Reviewer_PPjJ · 2023-11-22
>
> Thank you addressing my concerns. I have read authors response to all reviews. I would recommend authors to add the success rate in the main paper and results for testing generalization to unseen variants of basic split games. In addition, it would be nice if authors add more details to describe the combined baseline in the main paper, the current description is unclear. It'd also be good to add details about hyperparameters like max steps chosen for RL training in appendix. As most of my concerns are addressed, I will update my rating to reflect the score.

---

### Official Review · Reviewer_p9BE · 2023-10-29

**Soundness:** 3 good
**Presentation:** 3 good
**Contribution:** 2 fair
**Rating:** 6
**Confidence:** 3

**Summary:**

The paper presents a series of intricate tasks centered on physical reasoning and interaction. Additionally, it suggests three distinct approaches for task resolution using supervised or reinforcement learning methods. To conclude, an extensive user study is executed to gauge human proficiency across various learning techniques.

**Strengths:**

The study introduces a valuable benchmark for assessing model predictions concerning physical outcomes. The benchmark's interactive nature facilitates real-time planning, crucial for real-time physics interactions, particularly when timing plays a key role in the dynamics. Furthermore, it supports multi-step interventions, promoting long-term predictions over brief, single-step action forecasts.

**Weaknesses:**

1. Please add a table grid to Figure 2 for clearer comparisons.
2. The author notes the absence of 3D interactive environments as a limitation. However, this is a significant point to address since many high-performing models should ideally transfer seamlessly to robotics. A 3D interactive environment would greatly facilitate this transition. Notably, papers like ComPhy feature 3D environments, as referenced in Table 1 by the author.
3. I appreciate the author's use of this paper as a benchmark for various RL approaches, highlighting the need for more research in this domain to address physical reasoning tasks. What potential solutions or recommendations does the author suggest for this benchmark?

**Questions:**

Mentioned in weakness

---

> ### Author Response · Authors · 2023-11-19
> **Response to Reviewer p9BE**
>
> We are grateful for your recognition of I-PHYRE's importance and address your concerns below:
>
> > *Add a table*
>
> We will include a table in the revised version, space permitting.
>
> > *The author notes the absence of 3D interactive environments as a limitation.*
>
> While we acknowledge the potential benefits of a 3D interactive benchmark, our focus on 2D is a deliberate step from PHYRE's design, emphasizing interactivity. We hypothesize that the main challenge in 3D relates to perception, and with advancements in the vision field, a robust physics reasoning method will be increasingly valuable.
>
> > *What potential solutions or recommendations does the author suggest for this benchmark?*
>
> As outlined in Section 5.1, we propose focusing on physics modeling, multi-step interventions, and action timing to build more powerful physical reasoning agents. The [schema network (ICLR 17')](https://arxiv.org/pdf/1706.04317.pdf) is a promising starting point, advocating for transparent environment modeling to improve optimization and planning.

---

> > ### Comment · Reviewer_p9BE · 2023-11-22
> > **Response to rebuttal**
> >
> > I would like to thank the authors for their response, after consideration, i would keep my rating.

---

### Official Review · Reviewer_yzoQ · 2023-10-31

**Soundness:** 3 good
**Presentation:** 3 good
**Contribution:** 3 good
**Rating:** 8
**Confidence:** 4

**Summary:**

This paper proposes I-PHYRE, an interactive physical reasoning benchmark. While previous physical reasoning benchmarks mainly focus on reasoning happening in stationary scenes, I-PHYRE tests physical reasoning in an interactive format. This demands the agent to quickly understand the underlying physics, plan over multi-steps, and perform timely manipulation within a scene. The authors formulated I-PHYRE into four game splits to measure the agents' capability to learn and generalize about essential principles of physics, and
conducted extensive evaluations of existing learning algorithms and human performance. They showed a significant gap between human and learning algorithms, and also analyzed what are the main factors of current learning methods fails.

**Strengths:**

1. Clear motivation and novel design achieving the motivation.
     - The authors clearly stated I-PHYRE’s contribution compared to other existing benchmarks.
     - The proposed task of block elimination is well designed for the stated purpose, highlighting interactivity of physical reasoning.
     - Additionally, dividing the tasks into 4 types for testing different types of generalization is well designed.
2. Thorough experiments and analysis
     - The experiments are very thorough, including multiple reinforcement learning baselines, multiple planning strategies, learning from offline data, integration with large language model(LLM), human evaluation, etc.
     - The authors also tried to analyze what makes learning algorithms difficult to generalize, and came up with three plausible factors.
     - Additionally, the authors also quantitatively analyzed how significant action timing influences the performance of the algorithms.

**Weaknesses:**

1. The visualization of Figure 1. is slightly hard to understand at first glance. Although the authors explain the figure more thoroughly in page 3, it would be better to either separate the figure into several images or add another image that describes the task more briefly.
2. Although the experiment was thorough, it can be that 40 games is a small number of games to anticipate strong generalization. It would be better if the authors have tested learning methods in a larger scale. For instance, the authors can try more diverse configuration given the same basic physics to model in the scene.
3. The cited work [1] also contains interactivity in the benchmark, although it mainly tackles generalization to new actions as the authors mentioned.



[1] Generalization to New Actions in Reinforcement Learning, International Conference on Machine Learning, 2020, Jain et al.

**Questions:**

1. I wonder why the authors simply concatenated the predicted states with the current states when performing model based reinforcement learning. A more dominant approach to use would be
     1. Along with the dynamics model, also train a reward model that predicts the reward given a state.
     2. Given a reward model and a dynamics model, perform planning (e.g. CEM planning)

2. There are also other recent model based RL methods the authors can try, such as [1].



[1] Temporal Difference Learning for Model Predictive Control, International Conference on Machine Learning, 2023, Hansen et al.

---

> ### Author Response · Authors · 2023-11-19
> **Response to Reviewer yzoQ**
>
> Thank you for appreciating our clear motivation, novel design, and thorough experiments and analysis.
>
> > *The visualization of Figure 1. is slightly hard to understand at first glance.*
>
> We acknowledge your feedback. In response, we will divide Figure 1 into two subfigures to more effectively illustrate the challenges of I-PHYRE and the creation of game splits.
>
> > *Although the experiment was thorough, it can be that 40 games is a small number of games to anticipate strong generalization.*
>
> We appreciate your concern. We limited the scale of I-PHYRE for several reasons:
>
> 1. **Prior evidence against large-scale learning**: Despite a large dataset, methods excelling in within-template tasks struggle in cross-template settings (as seen in PHYRE).
> 2. **Minimal yet complete system**: Scaling up can introduce unsolvable games due to the need for precise timing. Our goal is to probe the boundary of intuitive physics models, and the current simulator effectively serves this purpose.
> 3. **Learning generalizable and compositional physics**: We focus on generalization from learning physical primitives and composing them, as supported by research like the [schema network (ICML 17')](https://arxiv.org/pdf/1706.04317.pdf), which generalizes from a single setup to a range of variations.
>
> > *Related work*
>
> We will extend our discussion on related work in the revised manuscript.
>
> > *More dominant model-based reinforcement learning.*
>
> We will explore recent methods in model-based reinforcement learning and integrate them into our discussion, following the World Model paradigm.

---

> ### Comment · Reviewer_yzoQ · 2023-11-23
> **Response to rebuttal**
>
> Thanks for your comments. I have read the response and most of my concerns are addressed. I will keep my current rating as 8.

---

### Official Review · Reviewer_BYPY · 2023-11-04

**Soundness:** 3 good
**Presentation:** 2 fair
**Contribution:** 3 good
**Rating:** 6
**Confidence:** 4

**Summary:**

This paper introduces I-PHYRE, a benchmark for intuitive physical reasoning capabilities in decision-making agents. It consists of four different block-removal games and benchmarks three planning strategies against them, implemented with both supervised and reinforcement learning. I-PHYRE's design centers on three principles: physical reasoning, multi-step planning, and in-situ intervention.

The four games are "basic" (teaching basic principles of physics), "noisy" (minor perturbations), "compositional" (combining various structures to require multi-step reasoning), and "multi-ball" (multiple dynamic events occurring concurrently, motivating carefully timed in-situ intervention). The planning strategies are "planning-in-advance" (generating an entire plan with timings based only on the initial state), "planning-on-the-fly" (generating the next action at each timestep given the observation - standard RL-type setup), and "combined" (generating the entire plan, then updating it after executing the first action).

Experiments include results from model-free deep RL agents as well as some other baselines in supplementary on the games. Everything is compared to a human baseline. The paper specifically discusses how agents perform when generalizing (without further training) from the basic split to other splits, as well as how the three training strategies differ.

Finally, the paper gives analysis of sources of difficulty in I-PHYRE, performance by offline algorithms, and limitations/future work.

**Strengths:**

#### Quality
- Overall, a solid paper. Intuitively, if an agent did well on I-PHYRE, I would believe that it had robust intuitive physics capabilities in certain domains, which is what a benchmark should convince me of.
- Appendix G is very valuable. It does rest on the assumption that all failures are either of bad order or bad timing, which eliminates more basic failures, but since it is a simple simulator, and since Appendix G clearly shows that all the failures in Basic are timing-based (i.e. all the failed plans have correct order), I am convinced that the Basic split teaches nontrivial principles and the other splits

#### Clarity
- Very well-written! Clear language
- The text organization is really useful, especially in the results section. The subsections make sense and each paragraph follows a claim-warrant structure. In terms of readability, papers often fall apart in the results section; this one does not.

#### Originality
I-PHYRE seems to have an original design. However, it's not the first interactive intuitive physics benchmark. The paper should compare to [1] and [2], though I do think it serves different, valuable purposes.

#### Significance
The tasks in I-PHYRE are distinct from other related work and test compelling aspects of intuitive physics reasoning. So, I think if the paper can prove its claims, it is significant.

[1] Physion: Evaluating Physical Prediction from Vision in Humans and Machines. Bear, D. M. et al. arxiv preprint: arXiv:2106.08261. 2021.
[2] Jain, A. et al. "Generalization to new actions in reinforcement learning." ICML 2020.

**Weaknesses:**

#### Quality
- The paper says that RL agents perform well on the noisy split, sort of justified by the fact that their noisy results "correlate" with their basic results. But that doesn't follow - the noisy results are clearly of lower magnitude even if they follow the same trends (across what factor of variation?), and we aren't given a correlation statistic to warrant this claim. The same applies to "correlation diminishes in the compositional and multi-ball splits... inherent complexities of these tasks impact performance negatively." The correlation claim is similarly 1) nonobvious from looking at Fig 2, even if it does sort of look true, 2) not backed up by a number, 3) not clear why it matters - even if the performance on these two were correlated with perf on basic across whatever factor of variation, but lower in magnitude, I would accept that they are harder (and I of course do, just from looking at Fig 2). Then making a claim that this difficulty is due to the "inherent complexities of these tasks" is, while not unbelievable, strong. I'm not doubtful, just not convinced.
- The discussion has a section titled "why do current RL agents fail on I-PHYRE?", which in my opinion is the most important aspect of a benchmark paper other than conceptual design and grounding in the environment. The paper claims three benefits: physics modeling being hard, multi-step interventions, and action timing - i.e. asserting that its design principles have effectively resulted in challenges for agents. However, this is all prose and little analysis of actual results - it would help to spell out for the reader which quantitative result comparison should lead to each conclusion. (Appendix G does a good job of this).

#### Clarity
- Figure 1 (repeated from a previous review I did of this paper, as the figure hasn't changed):
  - Colors are hard to follow, maybe better to annotate split on box
  - Since the boxes are much larger than the arrows, sort of look like two columns, and generally don't look like flowchart elements, it's confusing that the top-left box isn't the best one to start reading with. Bigger arrows and/or numbering would help.
  - "Wrong order leads to no overlap elimination timing for two balls" - I don't understand this
  - Compositional solution isn't that clear - annotation of key occurrences in each step might help
- "Combined" planning strategy needs better explanation (a step-by-step could help) - it sounds like the whole plan is generated based on the initial state, then the first action is executed, then the plan is updated, and that's it. But I'm not totally sure if the entire plan is updated, if it's ever updated again later after subsequent actions, etc.
- Figure 2 needs to be more organized and readable (in the previous version it was also hard to follow, but still more organized)
  - Strategies should be put into separate subplots (with the same axes) or otherwise cued, especially since there are different numbers of each, making it hard to eyeball
  - Baselines (especially the human comparison point) can be horizontal lines crossing the figure
  - Not every algorithm gets every planning strategy, which doesn't seem to *only* be a function of the nature of the algorithm - so it's confusing to take all of this in just in the form of bars and text
- Fig 3 would benefit from organization as well - e.g. group strategies by color and differentiate within them by line texture.

#### Originality
- The paper claims the three planning strategies as a contribution, but they aren't original - they are baselines just like the offline methods tested on I-PHYRE. I agree that the *results* are a contribution, but I would fold that into the third contribution bullet, or at least make it clear that the "devised" planning strategies themselves shouldn't be considered original.

Nits:
- Different parts of the paper use different naming conventions for agents - e.g. "SAC-I" in one place, "SAC Inadvance" in another. Better to use the same thing throughout.

**Questions:**

- In a previous version of the paper, failure on I-PHYRE was ascribed to sparse action requirements and delayed reward. Now, those concepts are being pitched as being inherent to multi-step reasoning (delayed reward) and action timing (sparse action requirements), but it's not true that failing for those reasons means that if the agents were better at handling them, they would robustly learn to handle multi-step reasoning problems and in-situ intervention problems. Could you flesh this argument out more?
- What exactly is the nature of the "combined" strategy, and could you say more on why it's effective?

---

> ### Author Response · Authors · 2023-11-19
> **Response to Reviewer BYPY**
>
> We are glad that you treat our paper as solid with both good intuitive and empirical results and acknowledge the contribution and significance of I-PHYRE. We will discuss and compare our work with [1] and [2] in the revised version.
>
> For your concerns:
>
> > *Then making a claim that this difficulty is due to the "inherent complexities of these tasks" is, while not unbelievable, strong.*
>
> We acknowledge the challenge in analyzing the innate complexities of games. Our research shows these complexities by highlighting the **sampling difficulty in identifying successful solutions** in Appendix B, providing an approximate estimation of game difficulties. We will clarify this point in the revised version.
>
> > *Failure reasons*
>
> We appreciate your comments! In the revised version, we will elaborate on the experimental results presented in Appendix G and strengthen our claims with empirical evidence and hypotheses.
>
> > *On Figure 1*
>
> Thank you for your suggestions. In response:
>
> 1. We used larger arrows to illustrate relationships and will enlarge them further if space permits.
> 2. We will add additional annotations and coloring.
> 3. We changed the caption to "Wrong order leads to missing elimination timing for two balls" and referred readers to our supplementary demo video for a detailed explanation.
>
> > *"Combined" planning strategy needs better explanation.*
>
> The plan updates after each action. For example, if the initial plan involves actions at times t1 and t2 (t1 < t2), the agent waits until t1, executes the action, and updates the plan for the following action.
>
> > *On Figure 2*
>
> We will revise Figure 2 to:
>
> 1. Separate the bars according to strategies.
> 2. Depict human results as horizontal lines for direct comparison.
>
> > *On Figure 3*
>
> We will organize Figure 3 into groups in the revised version.
>
> > *Contribution statements*
>
> We will incorporate your suggestions in the revised manuscript.
>
> > *Different naming conventions for agents*
>
> We will ensure consistency in agent naming throughout the paper.
>
> > *Failure on I-PHYRE*
>
> We view multi-step reasoning and action timing as **scientific** problems, while delayed reward and sparse action are **computational** challenges. The multi-step nature causes delayed rewards, and action timing implies sparsity. We will distinguish hypotheses from claims in the revised version.
>
> > *Nature of the "combined" strategy*
>
> The combined strategy, inspired by human thinking, maintains a global vector of pre-planned action timing, updating it intermittently. This approach balances between constant updates and pre-planning, re-planning only at key frames for computational efficiency and effectiveness.

---

### Meta-Review · Area_Chair_BzcP · 2023-12-03

**Metareview:**

The paper received positive ratings from the reviewers (three “marginally accept” and one “accept”). The reviewers had various concerns, for example, (1) lack of evidence for some of the claims, (2) planning strategies not being the contributions of this paper, (3) a small number of games to claim generalization, (4) being limited to 2D environments. The rebuttal addressed some of the concerns by the reviewers. Despite these weaknesses, the AC and the reviewers believe the paper provides an interesting benchmark for intuitive physics, and the experiments are thorough. Therefore, the AC follows the recommendation of the reviewers and recommends acceptance.

**Justification For Why Not Higher Score:**

The paper exhibits some weaknesses. For instance, it does not consider 3D environments, which are more realistic compared to the proposed benchmark. The authors' justification for using 2D environments is that they have followed prior work from four years ago, but this argument is not compelling.

**Justification For Why Not Lower Score:**

The paper proposes an interesting benchmark which helps advance the reasoning for intuitive physics.

---

### Decision · Program_Chairs · 2024-01-16

Accept (poster)